# Laparoscopic Intraperitoneal Onlay Mesh (IPOM): Short- and Long-Term Results in a Single Center

Mario Giuffrida *, Matteo Rossini, Lorenzo Pagliai, Paolo Del Rio and Federico Cozzani

General Surgery Unit, Department of Medicine and Surgery, Parma University Hospital, 43123 Parma, Italy
* Correspondence: mario.giuffrida4@gmail.com

**Abstract:** The laparoscopic intraperitoneal onlay mesh repair (IPOM) approach has become the most widely adopted technique in the last decade. The role of laparoscopic IPOM in the last years has been resizing due to several limitations. The aim of the present study is to evaluate short- and long-term outcomes in patients who underwent laparoscopic IPOM. This retrospective single-center study describes 170 patients who underwent laparoscopic IPOM for ventral hernia at the General Surgery Unit of Parma University Hospital from 1 January 2016 to 31 December 2020. We evaluated patient, hernia, surgical and postoperative characteristics. According to the defect size, we divided the patients into Group 1 ($\varnothing$ < 30 mm), Group 2 (30 < $\varnothing$ < 50 mm) and Group 3 ($\varnothing$ > 50 mm). A total of 167 patients were included. The mean defect diameter was 41.1 ± 16.3 mm. The mean operative time was different among the three groups ($p < 0.001$). Higher Charlson Comorbidity Index, obesity and incisional hernia were related to postoperative seroma and obesity alone with SSO. $p < 0.001$ Recurrence was significantly higher in larger defects (Group 3) and incisional hernia. $p < 0.001$. This retrospective study suggests that laparoscopic IPOM is a feasible and safe surgical technique with an acceptable complication rate, especially in the treatment of smaller defects up to 5 cm.

**Keywords:** hernia; mesh; ipom; abdominal; surgery; laparoscopy





## 1. Introduction

During the past decade, the treatment of ventral hernia has evolved due to the introduction of minimally invasive approaches [1–3].

Minimally invasive approaches for ventral hernia repair have shown several advantages over open techniques. Minimally invasive approaches enhance recovery, improve the quality of life and reduce overall health costs [4–6].

In this context, the laparoscopic intraperitoneal onlay mesh repair (IPOM) approach has become the most widely adopted technique.

IPOM repair was introduced by Karl Leblanc in 1993 [7].

IPOM repair consists of bridging the defect from the peritoneal side with a composite mesh. It is easy to perform and allows the placement of a larger mesh with minimal need for dissection.

This technique has offered several advantages compared with open repair in terms of lower risk of surgical-site occurrence (SSO), lower recurrence rate, better cosmetic outcome, shorter hospital stay and faster return to daily activities [8–11].

Despite the initial enthusiasm, IPOM techniques have shown several limitations. IPOM through fixating the mesh to the abdominal wall, commonly performed with tacks and/or transfascial sutures, has been linked to increased postoperative chronic pain and hematoma. Furthermore, the mesh is exposed to intra-peritoneal contents increasing the risk for the development of intra-peritoneal adhesions and mesh migration into the bowel due to excessive inflammatory reaction. Mesh development has been able to diminish this issue by changing the pore size and creating multi-layered meshes, but the problem of intra-peritoneal adhesions and mesh migration has not been fully resolved [12–17].

The significant incidence of postoperative bulging or eventration of mesh, seromas, recurrences and non-restoration of abdominal muscle function led to the introduction of a modified IPOM technique defined as the IPOM plus technique that consists of the sutured closure of the defect, which may restore function and prevent bulging [18–21].

Literature findings reported decreased recurrence and seroma rates in the IPOM plus group [22,23].

The introduction of new robotic techniques with extraperitoneal mesh placement (retromuscular or preperitoneal) and other new surgical techniques are responsible for the resizing of laparoscopic IPOM in the treatment of ventral hernia [3,24,25].

The 2020 EHS guidelines confirmed this trend suggesting laparoscopic hernia repair for defects larger than 4 cm and with a preperitoneal or retromuscular mesh positioning. In medium-sized hernia, the EHS suggest the laparoscopic approach in patients at high risk of wound infection, such as obese patients [26].

The EHS made this recommendation in favor of a new minimally invasive approach with a preperitoneal or retromuscular mesh positioning. However, as reported, the quality of evidence of these papers is poor such as the strength of recommendation is weak due to the lack of long-term studies.

Despite the several limitations of the IPOM technique—especially for larger defects—it is one of the easiest techniques for laparoscopic ventral hernia repair and is safe and reproducible.

The aim of the present study is to evaluate short- and long-term outcomes in patients who underwent laparoscopic IPOM in a single institution.

## 2. Methods

This retrospective single-center population-based study describes 170 patients who underwent laparoscopic intraperitoneal onlay mesh repair (IPOM) for ventral hernia at the General Surgery Unit of Parma University Hospital from 1 January 2016 to 31 December 2020, according to the STROBE statement. Informed consent was obtained from all participants.

Follow-up information was obtained via the electronic medical record for subsequent hospitalizations or clinic visits, and a questionnaire was administered to every patient from April 2022 to September 2022.

Inclusion criteria: Patients over 18 and patients with primary or recurrent ventral hernia treated with laparoscopic IPOM.

Exclusion criteria: Patients under 18, patients with ventral hernia treated with open approach or laparoscopically without IPOM technique, patients who underwent concomitant other surgical procedures, and patients treated with IPOM plus technique because only two times has been performed in the period of analysis.

Before surgery, every patient was evaluated through physical examination and, if necessary, with abdominal ultrasound and/or CT scan of the abdomen without c.e to detect and define hernia characteristics in order to plan the right treatment choice.

*Surgical Technique*

The procedure was performed under general anesthesia. The patient was in the supine position with both arms at 90°.

Pneumoperitoneum at 12 mm Hg was established using either a Veress needle or an "open technique" according to the surgeon's preferences. The first 10–12 mm trocar was inserted laterally between the anterior iliac spine and the subcostal margin (anterior axillary line). Two working trocars were placed under vision, the first one of 5 mm and the second of 10–12 mm.

Adherences were grasped, and adhesiolysis was carried out, preferably with cold scissors, to clear the margins of the defect and to avoid bowel injury. After adhesiolysis was performed, a reduction of hernia contents was started with the steady hand-over-hand withdrawal of the sac contents. The reduced part of the viscera was inspected to rule out ongoing bleeding. The peritoneum flap was always created using monopolar diathermy.

Different meshes have been used through the years.

The defect size was measured for each side, and then the prosthetic mesh was tailored to ensure at least a 5 cm overlap of all defect margins.

Plenty of different types of mesh are available in the market. We used several different types of meshes.

Absorbable and non-absorbable tacking devices were always used for mesh fixation with the double-crown technique. During mesh fixation, the pneumoperitoneum pressure was always reduced at 8–10 mm Hg

Data included patient demographics (age, gender, BMI, comorbidities, history of smoking, diabetes, cardiopulmonary diseases, high blood pressure, COPD, chronic renal failure, chronic use of steroids or immunosuppressant drugs, and Charlson Comorbidity Index); hernia characteristics (previous abdominal surgery, diagnostic exams, abdominal ultrasound or CT scan of the abdomen, hernia site, hernia width, number of defects according to EHS classification) [27]; surgical details (type of mesh, mesh size, mesh fixation system, operative time, intraoperative complications); postoperative course (length of stay, Clavien-Dindo Classification, Surgical Site Infection—SSI, Surgical Site Occurrence—SSO, postoperative pain); and follow-up (chronic pain, mesh bulging, SSI, SSO, recurrence, recurrence treatment).

We divided patients into three main groups according to the defect size.

Group 1 patients with a defect smaller than 3 cm width, Group 2 with a defect width between 3 and 5 cm and Group 3 with a defect larger than 5 cm width.

The endpoints of this study were the analysis of short- and long-term outcomes (SSI-SSO-recurrence rate-chronic pain) evaluation of the possible factors related to the worst outcomes.

Data analysis was performed using JASP and JAMOVI.

Statistical analysis was obtained for the main descriptive indexes.

Quantitative data are expressed as mean or median ± standard deviation (SD) or median and interquartile range (IQR, minimum and maximum values). The qualitative data were elaborated as absolute frequencies, relative frequencies, cumulated frequencies and percentages.

Factors that are clinically relevant based on literature and/or expert opinion have been selected for multivariate analysis.

Student's *t* test or ANOVA were used for comparisons of continuous variables between groups. Chi-squared test or Fisher's exact test, as appropriate, is used for the analysis of categorical data.

All factors were deemed to be statically significant for a *p*-value of less than 5% ($p < 0.05$).

## 3. Results

A total of 167 patients who underwent laparoscopic IPOM were included. Two patients were excluded because they underwent ventral hernia repair with IPOM plus technique. The mean age was 58.7 ± 12.6 years. There were 79 male (47.3%) patients and 88 female (52.7%) patients.

### 3.1. Baseline Characteristics

Age distribution was similar in the three groups. Gender distribution was similar in the second and third groups, with a little prevalence of males in Group 1.

Charlson Comorbidity Index score was non significantly higher in Groups 2–3 (2.3 ± 1.8 and 2.5 ± 2.3, respectively, in Groups 2 and 3 versus 1.7 ± 1.5 of Group 1) than in Group 1. 96% of patients in Group 3 underwent previous abdominal surgery against the 76.3% of patients in Group 1.

ASA score distribution showed a similar distribution among the three groups.

Patient characteristics are summarized in Table 1.

**Table 1.** Baseline characteristics.

| Baseline Characteristics | Total | Group 1 (Ø < 3 cm) | Group 2 (3 < Ø < 5 cm) | Group 3 (Ø >5 cm) | *p*-Value |
|---|---|---|---|---|---|
| Median age, mean (SD) | 58.7 ± 12.6 | 57.6 ± 10.9 | 59.9 ± 12.9 | 57.3 ± 14.8 | 0.965 |
| Male sex, *n* (%) | 79 (47.3%) | 33 (60.0%) | 36 (41.3%) | 10 (40.0%) | 0.659 |
| BMI > 25, *n* (%) | 63 (37.7%) | 19 (34.5%) | 31 (35.6%) | 13 (52.0%) | 0.418 |
| Smokers, *n* (%) | 13 (7.7%) | 6 (10.9%) | 5 (5.7%) | 2 (8.0%) | 0.928 |
| Type II Diabetes, *n* (%) | 13 (7.7%) | 3 (5.4%) | 9 (10.3%) | 1 (4.0%) | 0.423 |
| Hypertension, *n* (%) | 53 (31.7%) | 16 (29.0%) | 26 (29.8%) | 11 (44.0%) | 0.483 |
| Ischaemic Heart Disease, *n* (%) | 10 (5.9%) | 1 (1.8%) | 8 (9.1%) | 1 (4.0%) | 0.338 |
| COPD, *n* (%) | 4 (2.3%) | 1 (1.8%) | 3 (3.4%) | 0 (0.0%) | 0.539 |
| Chronic Kidney Disease stage III, *n* (%) | 5 (2.9%) | 0 (0.0%) | 4 (4.5%) | 1 (4.0%) | 0.763 |
| Charlson Comorbidity Index, mean (SD) | 2.1 ± 1.8 | 1.7 ± 1.5 | 2.3 ± 1.8 | 2.5 ± 2.3 | 0.858 |
| ASA score | | | | | 0.121 |
| 1 | 27 (16.1%) | 6 (10.9%) | 17 (19.5%) | 4 (16.0%) | |
| 2 | 79 (47.3%) | 24 (43.6%) | 44 (50.5%) | 11 (44.0%) | |
| 3 | 59 (35.3%) | 25 (45.4%) | 25 (28.7%) | 9 (36.0%) | |
| 4 | 2 (1.1%) | 0 (0.0%) | 1 (1.1%) | 1 (4.0%) | |
| Previous abdominal surgery, *n* (%) | 141 (84.4%) | 42 (76.3%) | 75 (86.2%) | 24 (96.0%) | 0.803 |

### 3.2. Hernia Characteristics

Primary hernia was the most common hernia in 80 (47.9%) cases.

CT scan of the abdomen was performed in 55 (32.9%) patients and ultrasound in 22 (13.1%) cases. The mean defect diameter was 41.1 ± 16.3 mm (range 17–120 mm). Intraoperatively, additional defects were identified in 12 cases (7%).

A total of 55 (32.9%) patients were in Group 1, 87 (52.0%) in Group 2 and 25 (14.9%) in Group 3.

Umbilical hernia was the most common in 71 patients (42.5%). This type of hernia had a similar distribution among the three groups. According to the EHS classification, M3 W1 was the most common hernia in 55 cases (52.8%).

The hernia characteristics are summarized in Tables 2 and 3.

**Table 2.** Hernia characteristics.

| Hernia Characteristics | Total | Group 1 (Ø < 3 cm) | Group 2 (3 < Ø < 5 cm) | Group 3 (Ø > 5 cm) | *p*-Value |
|---|---|---|---|---|---|
| Hernia type, *n* (%) | | | | | |
| Primary hernia | 80 (47.9%) | 32 (58.1%) | 38 (43.6%) | 10 (40.0%) | 0.964 |
| Incisional hernia | 70 (41.9%) | 20 (36.3%) | 39 (44.8%) | 11 (44.0%) | |
| Recurrent incisional hernia | 17 (10.1%) | 3 (5.4%) | 10 (11.4%) | 4 (16.0%) | |
| Diagnostic test, *n* (%) | | | | | |
| Ultrasound | 22 (13.1%) | 9 (16.3%) | 11 (12.6%) | 2 (8.0%) | 0.765 |
| CT scan | 55 (32.9%) | 10 (18.1%) | 35 (40.2%) | 10 (40.0%) | |
| Defect location, *n* (%) | | | | | |
| Umbilical hernia | 71 (42.5%) | 29 (52.7%) | 34 (39.0%) | 8 (32.0%) | |
| Supraumbilical hernia | 28 (16.7%) | 8 (14.5%) | 15 (17.2%) | 5 (20.0%) | |
| Spigelian hernia | 5 (2.9%) | 1 (1.8%) | 2 (2.2%) | 2 (8.0%) | 0.985 |
| Lumbocele | 14 (8.3%) | 3 (5.4%) | 9 (10.3%) | 2 (8.0%) | |
| Median incisional hernia | 32 (19.1%) | 9 (16.3%) | 18 (20.6%) | 5 (20.0%) | |
| Other | 17 (10.1%) | 5 (9.0%) | 9 (10.3%) | 3 (12.0%) | |
| Number of defects > 2, *n* (%) | 19 (11.3%) | 6 (10.9%) | 9 (10.3%) | 4 (16.0%) | 0.342 |
| Defect size Ø mm, mean (SD) | 41.1 ± 16.3 | 28.4 ± 10.0 | 41.1 ± 5.9 | 68.7 ± 18.4 | <0.001 |

**Table 3.** EHS classification of hernia.

| Hernia Type *n* (%) | Total | W 1 (Ø < 4 cm) | W 2 (4 < Ø < 10 cm) | W 3 (Ø > 10 cm) | *p*-Value |
|---|---|---|---|---|---|
| M2 | 37 (22.1%) | 24 (23.0%) | 13 (23.6%) | 0 (0.0%) | 0.138 |
| M3 | 88 (52.6%) | 55 (52.8%) | 31 (56.3%) | 2 (25.0%) | |
| M4 | 4 (2.3%) | 1 (0.9%) | 2 (3.6%) | 1 (12.5%) | |
| M5 | 7 (4.1%) | 4 (38%) | 2 (3.6%) | 1 (12.5%) | |
| L1 | 5 (2.9%) | 3 (2.8%) | 1 (1.8%) | 1 (12.5%) | |
| L2 | 7 (4.1%) | 6 (5.7%) | 0 (0.0%) | 1 (12.5%) | |
| L3 | 14 (8.3%) | 10 (9.6%) | 3 (5.4%) | 1 (12.5%) | |
| Spigelian hernia | 5 (2.9%) | 1 (0.9%) | 3 (5.4%) | 1 (12.5%) | |
| Recurrent incisional Hernia | 17 (10.1%) | 2 (25.0%) | 5 (62.5%) | 1 (12.5%) | |

### 3.3. Intraoperative Details

The most common meshes were Bard Ventralight ST® Mesh in 114 (68.2%) patients and GORE® DUALMESH® in 45 (26.9%) patients.

Among the three groups, the Bard Ventralight ST® Mesh sizes were significantly different. ($p$ = 0.003)

A double-crown with non-absorbable fixation devices was always performed. No differences among fixation systems were found among the three groups. ProTack™ fixation system was used in 69 (41.3%) patients and CapSure™ fixation system in 68 (40.7%) patients.

Conversion to open surgery was not required among the 167 patients.

The mean operative time was different among the three groups ($p$ < 0.001). The mean operating time was slightly longer in group 3 (131.4 ± 76.7 min in Group 3, 76.4 ± 40.5 and 94.5 ± 55.2, respectively, in Groups 2 and 3). Intraoperative details are reported in Table 4.

**Table 4.** Surgical details.

| Surgical Details | Total | Group 1 (Ø < 3 cm) | Group 2 (3 < Ø < 5 cm) | Group 3 (Ø > 5 cm) | *p*-Value |
|---|---|---|---|---|---|
| Mesh type, *n* (%) | | | | | |
| GORE® DUALMESH® | 45 (26.9%) | 11(20.0%) | 28 (32.1%) | 6 (24.0%) | 0.964 |
| PROCEED® Surgical Mesh | 2 (1.1%) | 2 (3.6%) | 0 (0.0%) | 0 (0.0%) | |
| Ventralight™ ST Mesh | 114 (68.2%) | 40 (72.7%) | 57 (65.5%) | 17 (68.0%) | |
| Phasix™ Mesh | 1 (0.5%) | 1 (1.8%) | 0 (0.0%) | 0 (0.0%) | |
| Symbotex™ Composite Mesh | 5 (2.9%) | 1 (1.8%) | 2 (2.2%) | 2 (8.0%) | |
| Most common mesh size (cm), *n* (%) | | | | | |
| GORE® DUALMESH® | (15 × 19), 15 (33.3%) | (10 × 15), 6 (54.5%) | (24 × 18), 10 (35.7%) | (24 × 18), 4 (66.6%) | 0.901 |
| PROCEED® Surgical Mesh | (10 × 15), 2 (100%) | (10 × 15), 2 (100%) | / | / | / |
| Ventralight™ ST Mesh | (15.2 Ø), 39 (34.2%) | (15.2 Ø), 15 (62.5%) | (15.2 × 20.3), 24 (42.1%) | (18 × 23), 7 (41.1%) | 0.003 |
| Phasix™ Mesh | (20 × 25), 1 (100%) | (20 × 25), 1 (100%) | / | / | / |
| Symbotex™ Composite Mesh | (15 × 20), 3 (60.0%) | (15 × 20), 1 (100%) | (15 × 20), 2 (100%) | (28 × 37), 1 (100%) | 0.078 |
| Fixation system, *n* (%) | | | | | |
| CapSure™ | 68 (40.7%) | 24 (43.6%) | 32 (36.7%) | 12 (48.0%) | 0.777 |
| ProTack™ | 69 (41.3%) | 24 (43.6%) | 37 (42.5%) | 8 (32.0%) | |
| Both | 30 (17.9%) | 7 (12.7%) | 18 (20.6%) | 5 (20.0%) | |
| Operative time, mean (SD) | 93.9 ± 57.2 | 76.4 ± 40.5 | 94.5 ± 55.2 | 131.4 ± 76.7 | <0.001 |

In our study, we had a similar distribution of laparoscopic IPOM through the years, with a peak in 2019 with 47 (28.1%) surgical procedures and a mean of 33.4 procedures per year. The pandemic Sars-CoV-2 affected surgical procedures, especially for benign diseases, during 2020, in fact only 12 (7.1%) laparoscopic IPOMs were performed.

### 3.4. Postoperative Outcomes

The mean LOS was $3.31 \pm 2.6$ days (range 1–25 days). LOS was significantly different among the three groups. $p < 0.001$.

Complications within 30 days after surgery were reported in 7 (4.1%) patients. Severe complications (Clavien-Dindo > 3b) were reported in 1 (0.5%) patient who required surgery for small bowel perforation. No 90-day mortality has been reported.

The mean follow-up was $112.0 \pm 35.8$ months (range 9.3–176.7 months). Postoperative outcomes are summarized in Table 5.

**Table 5.** Postoperative outcomes and long-term outcomes.

| Surgical Details | Total | Group 1 (Ø < 3 cm) | Group 2 (3 < Ø < 5 cm) | Group 3 (Ø > 5 cm) | *p*-Value |
|---|---|---|---|---|---|
| LOS (days), mean (SD) | $3.1 \pm 2.6$ | $2.3 \pm 1.2$ | $3.1 \pm 2.0$ | $5.0 \pm 5.0$ | <0.001 |
| Complications within 30 days, *n* (%) | 7 (4.1%) | 2 (3.6%) | 3 (3.4%) | 2 (8.0%) | 0.686 |
| Complications after 30 days, *n* (%) | 14 (8.2%) | 2 (3.6%) | 8 (9.2%) | 4 (16.0%) | 0.169 |
| Follow-up (months), mean (SD) | $112.0 \pm 35.8$ | $92.6 \pm 32.4$ | $125.6 \pm 31.7$ | $107.8 \pm 36.9$ | <0.001 |
| SSI, *n* (%) | 7 (4.1%) | 1 (1.8%) | 6 (6.8%) | 0 (0.0%) | 0.383 |
| SSO, *n* (%) | 20 (11.9%) | 5 (9.0%) | 13 (14.9%) | 2 (8.0%) | 0.469 |
| Recurrence, *n* (%) | 11 (6.5%) | 1 (1.8%) | 6 (6.8%) | 4 (16.0%) | 0.019 |
| Chronic pain, *n* (%) | 6 (3.5%) | 2 (3.6%) | 2 (2.2%) | 2 (8.0%) | 0.407 |
| Seroma, *n* (%) | 14 (8.2%) | 3 (5.4%) | 9 (10.3%) | 2 (8.0%) | 0.595 |
| Mesh bulging, *n* (%) | 10 (5.9%) | 0 (0.0%) | 8 (9.1%) | 2 (8.0%) | 0.347 |

SSI was reported in 7 (4.1%), and SSO in 20 (11.9%). Recurrence was reported in 11 (6.5%) patients, with differences among the three groups. $p = 0.019$. No differences among the three groups were discovered for the other outcomes. $p > 0.05$.

During long-term follow-up, major complications associated with laparoscopic IPOM technique, such as mesh migration or enteric fistulas, have not been reported.

Multivariate analysis showed a significant correlation between patients with higher Charlson Comorbidity Index and postoperative seroma. $p < 0.001$.

Obesity has been related to postoperative seroma and SSO. $p < 0.001$.

Recurrence was significantly related to Group 2, especially with Group 3. $p < 0.001$.

Incisional hernias showed a significantly higher rate of seroma and recurrence than primary hernias. $p < 0.001$.

Among the different used meshes, no differences were found in postoperative outcomes. $p > 0.05$.

## 4. Discussion

At the start of the last decade, laparoscopic IPOM repair appeared to be a promising new standard for the treatment of ventral hernia due to less invasiveness and better recovery from surgery. Through the years, many concerns about this technique have risen due to the inefficacy of larger defects treatment and consequent higher recurrence rate.

In this retrospective single-center study, we recorded a relatively low rate of adverse events: SSI 4.1%; SSO 11.9%; recurrence 6.5%; chronic pain 3.5%; seroma 8.2%; mesh bulging 5.9%. We found a higher recurrence rate in patients with large defects and incisional hernias, as reported in other literature findings [28,29].

In our study, obesity was an independent risk factor for postoperative seroma and SSO [30,31].

The literature findings report a seroma rate of up to 12.2%; in our study, the seroma rate was 8.2% and was related to comorbidities, especially with obesity [32,33].

The hernia recurrence was higher in Group 3, where 16% of patients developed recurrence. Of the four patients developing hernia recurrence in this group, we think the cause of recurrence was the small mesh overlap, too small to cover the whole defect.

A higher mesh-to-defect-area ratio has been reported in previous literature as an additional potential factor for a lower risk of recurrence, confirming our poorer results in Group 3, where larger defects were related to a higher recurrence rate. Our results confirmed literature findings regarding poor outcomes for patients with defects larger than 10 cm [34,35].

The recurrence rate in Groups 1 and 2 were more acceptable, with only 4.9% of recurrence. This finding can be related to better surgical indications according to lower defect size.

Organ injuries who are not detected intraoperatively are an important complication that can occur during laparoscopic IPOM. Intra-abdominal adhesion is one of the most crucial findings of laparoscopic IPOM. Adhesiolysis may increase the risk of organ injury, especially bowel injuries. In our series, only one patient (0.5%) has experienced bowel perforation intraoperatively not detected and required surgery. The risk of organ lesion during adhesiolysis has been reported in 1–6% of laparoscopic IPOM repair cases [36,37].

Another concern with laparoscopic IPOM is chronic pain. It has been defined as a pain that lasts more than 3 months. This complication has been reported in 2–9.5% of cases of laparoscopic IPOM repair. It has been related to fixation systems, especially non-absorbable ones. In our series, it has been observed in 3.5% of cases using only non-absorbable fixation devices [38,39].

Another typical concern of laparoscopic IPOM was the bulging of the mesh. Traditionally, the development of bulging was considered the central nonfunctioning portion of the abdominal wall that can protrude into the hernial sac due to intra-abdominal pressure, which can be explained by Laplace's law [40]. In this study, mesh bulging was reported in 5.9% of patients and only in patients with larger defects.

This study has several limitations. First, since this was a retrospective review, the data collected from the local medical record system depended on the limited information provided by the surgeon performing the procedure and the physician on-call and varied in detail and reliability. Second, the study was not double-blinded, randomized, or controlled and was thus subject to bias and possible confounding factors that may have influenced the results.

A further limitation represents the use of different meshes and different mesh fixation.

The study has not included IPOM plus as a treatment due to the recent introduction to clinical practice in our institution. Fascial closure has been recommended to prevent the bulging of the abdominal wall and seromas after bridging repair in laparoscopic IPOM repair. The closure of the fascial defect is technically feasible, but evidence of any superiority over standard IPOM procedure is lacking. Laparoscopic IPOM plus is not practicable in larger hernial defects where excess wall tension leads to unsatisfactory results due to tension in the stitches, tears in the tissues, and considerable postoperative pain [18,41,42].

In conclusion, the findings of this retrospective study suggest that laparoscopic IPOM is a feasible and safe surgical technique with good postoperative outcomes and an acceptable complication rate. Our results suggest that laparoscopic IPOM may be preferred in smaller hernias up to 5 cm in diameter. In larger hernias, other surgical techniques should be considered to overcome a higher complication rate, as suggested by the EHS guidelines.

EHS suggest the adoption of eTEP over IPOM in the treatment of ventral hernia, but the literature findings are promising but too young to justify the theoretical advantages over the traditional IPOM technique, leading to the abandonment of this technique [43–45].

We believe that the IPOM technique is an easier technique compared to eTEP or endoscopic/mini-open sublay (eMILOS) approach, and it could be considered an important treatment option in small defects lower than 5 cm.

**Author Contributions:** All authors participated equally in this research and preparation of the manuscript. Conceptualization, M.G. and M.R.; methodology, M.G.; validation, L.P., F.C. and M.R.; formal analysis, M.G.; data curation, F.C.; writing—original draft preparation, L.P.; writing—review and editing, M.G.; visualization, P.D.R.; supervision, P.D.R.; project administration, M.G. All authors have read and agreed to the published version of the manuscript.

**Funding:** The authors received no financial support for the research, authorship, and/or publication of this article.

**Institutional Review Board Statement:** The study has been approved by our institution's independent ethics committee (Comitato etico AVEN—area vasta Emilia nord, n° 30426—19 November 2021).

**Informed Consent Statement:** Written informed consent has been obtained from the patients to publish this paper.

**Data Availability Statement:** Data is contained within the article.

**Conflicts of Interest:** The authors declare no competing interest.

## Abbreviations

| | |
|---|---|
| AWR | Abdominal wall repair |
| CDC | Centers for Disease Control |
| SSI | Surgical Site Infection |
| SSO | Surgical Site Occurrence |

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
