# Peer review of "Laparoscopic Intraperitoneal Onlay Mesh (IPOM): Short- and Long-Term Results in a Single Center"

_2673-4095, doi:10.3390/surgeries4010011_

Round 1

Reviewer 1 Report

Thank you for giving  me a chance to review this article. Currently, IPOM is well-established, on the other hand, EHS have already published a guidelines for abdominal hernia with recommendation of eTEP. Current manuscript does not reach the level of publication.

1. Too many paragrapghs make difficult to read.

2.  The classification of hernia orifice should be also provided by EHS classification. 

3. Did you employ IPOM-plus procedure in small hernia?

4. As I mentioned, IPOM and IPOM-plus have been established, therefore, you should discuss about eTEP. And, if you have some cases of eTEP, please compare the results.

Reviewer 2 Report

Congratulations on a well presented surgical experience and a sound scientific method.

Two minor suggestions:

1.In the Era of robotic repairs it would be appropriate to give a mention of this approach.

2. there is a typo in the LOS 33days I  believe is 3.3 days

Round 2

Reviewer 1 Report

Thank you for giving me a chance to review this article. The authors have revised their article along with my suggestion.